# Untargeted metabolomics for the early detection of preeclampsia: A systematic review of human studies

Celia García-Mañas[1], María Dolores Jara Montes[2], Ana Cristina Abreu[1], Manuel A. Rodríguez Maresca[2], Ignacio Fernández[1]*, Ana Maria Fernandez Alonso[2]*

1 Department of Chemistry and Physics, Research Centre CIAIMBITAL, University of Almería, Sacramento, Almería, Spain, 2 Torrecárdenas University Hospital, Almería, C. Hermandad de Donantes de Sangre, Spain

* ifernan@ual.es (IF); anafernandez.alonso@gmail.com (AMFA)

## Abstract

This systematic review synthesizes current evidence on metabolomics-based biomarkers for the early prediction or diagnosis of preeclampsia and highlights promising candidates with potential clinical application. Following PRISMA guidelines, a comprehensive search was performed in PubMed, Cochrane Library, Web of Science, and ClinicalTrials.gov up to September 2024, using predefined terms related to preeclampsia, metabolomics, and pregnancy. Study selection and risk of bias assessment were conducted with CADIMA, applying PICO-based inclusion criteria and predefined quality appraisal standards. Of 112 records identified, 16 were duplicates, 57 were excluded after title and abstract screening, and 31 after full-text review, leaving 12 studies for inclusion. These comprised cohort, case–cohort, case–control, validation, prospective control, and translational designs. Data extraction captured study characteristics, populations, methodologies, biological matrices, and main findings. Considerable heterogeneity was observed across studies, with limited overlap in identified metabolites. Nonetheless, alanine was reported in serum, lactate was observed in both serum and urine, and glutamate and glutamine were detected across serum, plasma, and placental tissue. These metabolites, interconnected through the Cori and glucose–alanine cycles, have been linked to hepatic dysfunction, immune regulation, and excitotoxicity. Overall, metabolomics shows strong potential as a sensitive tool for biomarker discovery in preeclampsia, though further research is required to confirm findings, improve reproducibility, and integrate metabolomic data with clinical parameters to support personalized medicine approaches.

## Systematic review registration

PROSPERO, CRD42024540619.

**Data availability statement:** All relevant data are within the paper and its Supporting Information files.

**Funding:** C. G-M. thanks Agencia Española Contra el Cáncer (AECC) for a predoctoral grant (PRDAM234245GARC). This work was supported by the Chair "Reference Laboratory in NMR-based Biomarkers" at the University of Almería (Spain).

**Competing interests:** The authors have declared that no competing interests exist.

## 1. Introduction

Hypertensive disorders of pregnancy (HDP) are a frequent pregnancy complication that endangers both women and their fetuses, leading to further consequences and long-term health issues [1]. These disorders include preeclampsia (PE) and eclampsia, chronic hypertension, chronic hypertension with superimposed PE and gestational hypertension (GH) [2]. PE represents up to 14% of maternal mortality and leads to between 10% and 25% of perinatal deaths [3]. It involves sustained high systolic/diastolic blood pressure (≥ 140/90 mm Hg) as well as proteinuria (≥ 300 mg/24 h) after 20 weeks of gestation in previously normotensive women. This condition is linked to maternal organ dysfunction such as acute renal insufficiency, liver, neurological or haematological complications, uteroplacental dysfunction, fetal growth restriction/intrauterine growth restriction and intrauterine death [4]. PE comprises two types: early-onset PE (EO-PE), which has a worse impact in the placenta and fetus and presents before 34 weeks of gestation, and late-onset PE (LO-PE), whose phenotype is less severe and presents after 34 weeks of gestation [5].

The pathophysiology of PE remains unknown, although it is believed that poor placental blood flow could be caused by maternal, fetal and placental factors and leads to hypertension and other PE symptoms [6]. Controlling blood pressure with antihypertensives, taking low-dose aspirin for prevention when indicated are some of the ways of managing PE. The definitive treatment would be delivery [7]. Even though PE was identified over 15 centuries ago, we still have not developed reliable methods to predict, because the model based in the combination of maternal characteristics, medical history, mean arterial pressure, uterine artery pulsatility index (UtAPI), and serum placental growth factor (PlGF) predicts less than 90% of EO-PE cases and 75% of preterm PE cases [8]. Other commonly used biomarkers for preeclampsia (PE), such as soluble fms-like tyrosine kinase-1 (sFlt-1) and pregnancy-associated plasma protein-A (PAPP-A), reflect placental dysfunction and angiogenic imbalance and are widely used in clinical risk assessment. While the sFlt-1/PlGF ratio is particularly valuable to rule out PE in the short term, its predictive performance can vary depending on cut-offs, study population, and timing, and further evidence is needed to confirm improvements in clinical outcomes is still limited [9,10]. PAPP-A has been evaluated in first-trimester screening for preterm PE, but its predictive performance is limited compared with PlGF and adds little when used alone in risk models [11]. Therefore, there is a need to seek new methods capable of predicting PE early. "Omics" sciences in the past years have been widely employed in the biomedical research field. These sciences include genomics, transcriptomics, proteomics or metabolomics. Regarding PE, numerous new pathways and factors related to its cause have been identified using omics technologies [12].

Metabolites function as direct indicators of biochemical activity. They consist of low molecular weight molecules (usually below 1500 Da) chemically transformed in metabolic processes, providing a functional insight into the cellular state [13]. Clinical metabolomics, although it is still an emerging high-throughput technique, has become a crucial tool for its potential to uncover new biomarkers, investigating metabolic pathways, categorizing patients, and revealing fundamental metabolic changes in the onset, progression, or response to diseases and therapeutic treatments [14,15].

Some examples of these are the recent metabolomics studies in metastatic colorectal cancer or COVID-19. In colorectal cancer, metabolites like lactate, glutamate and pyruvate are linked to progression-free survival [16]. In COVID-19, serum panels, including isoleucine, TMAO, glucose, and other metabolites, effectively predict disease severity and mortality [17].

Studying the metabolome can clear up pathophysiological mechanisms and provide vital information for precision medicine, employing non-invasive methods [18]. Different established techniques are widely applied in metabolomics: nuclear magnetic resonance (NMR) and mass spectrometry (MS), the latter coupled with gas or liquid chromatography, capillary electrophoresis, or ultra-performance liquid chromatography (UPLC) [19,20].

The aim of this study is to systematically review the existing literature on the application of metabolomics in predicting PE and report relevant statistical data and metabolomic parameters, such as differential biomarkers or variables influencing the potential predictability of the disease.

In this context, it is important to highlight that the metabolomic and metagenomic dynamics in pregnant women are particularly pronounced compared to other physiological conditions. This is due to the profound hormonal fluctuations and the extensive cardiocirculatory adaptations, especially those associated with the placental "vascular resistance." Such heightened dynamics may actually represent an opportunity, potentially enhancing the early detectability of biomarkers, which are expected to evolve into digital biomarkers in the near future.

## 2. Sources

This review was pre-registered in the International Prospective Register of Systematic Reviews (PROSPERO) under registration number CRD42024540619 and was conducted in accordance with the Preferred Reporting Items for Systematic Reviews and Meta-Analyses (PRISMA) guidelines [21].

Relevant literature was retrieved from three major databases: PubMed (including MEDLINE), the Cochrane Library, and Web of Science. The search strategy combined the following terms: ("preeclampsia" OR "pre-eclampsia" OR "eclampsia") AND ("metabolomics" OR "metabonomics") AND ("pregnancy" OR "gestational" AND "hypertension"). No filters were applied except in Cochrane, where the search was restricted to Title, Abstract, and Keyword fields, and in PubMed and Web of Science, where results were limited to human studies. In addition, ClinicalTrials.gov was queried using the terms ("preeclampsia" OR "pre-eclampsia" OR "eclampsia") AND ("metabolomics" OR "metabonomics"). This yielded six studies, all of which were either ongoing or had incomplete status: four were listed as "recruiting" and two had an "unknown" status, with no posted results available. Eligible studies included case-control, observational, and cohort designs that reported metabolomics data from pregnant women with and without PE. The review objectives and detailed research questions are provided in S1 Appendix.

### 2.1. Study selection

Study selection was conducted using the free web-based software CADIMA (https://www.cadima.info/index.php). Search results from the three databases were imported into the platform, which automatically merged the records and removed duplicates. Two reviewers independently screened titles, abstracts, and full texts, applying the predefined inclusion criteria based on the PICO framework (detailed in S2 Appendix). The PICO criteria were defined as follows: Population (P): Pregnant women at risk of or diagnosed with PE; Intervention (I): Use of metabolomic profiling or analysis to identify biomarkers; Comparator (C): Healthy pregnant women; Outcome (O): Identification and/or validation of metabolomic biomarkers for the prediction or diagnosis of PE. Studies not meeting these criteria were excluded.

After completing the selection process, a standardized data extraction table was developed to capture key variables relevant to the research objectives. These included general study information (authors, year, country), characteristics of the study population (maternal age, gestational age, follow-up time), methodological details, type of biological matrix analyzed, and main findings. The final dataset, once verified and consolidated by two reviewers, was exported from CADIMA

in Excel format for synthesis. All studies that employed metabolomic approaches to investigate PE were included, and for comparison purposes, studies were grouped according to the type of biological matrix analyzed.

To evaluate study quality and risk of bias, a critical appraisal was carried out within CADIMA (S1 Table), using pre-defined criteria adapted to the study designs and research context. These criteria focused on methodological rigor, sample size, and measurement validity. Three reviewers independently assessed each study, and any disagreements were resolved by consensus. The results of the quality assessment are graphically summarized in S1 Fig.

Due to substantial heterogeneity across studies—particularly in sample types, analytical platforms, metabolites assessed, and outcome definitions—a meta-analysis was not performed. Instead, a narrative synthesis was conducted, organizing findings by biological matrix to facilitate interpretation. Given the small number of included studies (n = 12) and the absence of a quantitative summary, subgroup and sensitivity analyses were not considered appropriate.

## 3. Results

A total of twelve studies met the inclusion criteria. The study selection process is illustrated in the PRISMA flow diagram (Fig 1).

A list of excluded studies, along with reasons for exclusion during the screening and eligibility phases, is provided in S2 and S3 Tables.

### Study characteristics

The selected studies were published between 2010 and 2024 (Table 1). They were conducted in various countries: United States [22–24], United Kingdom [25–27], Norway [28–30], China [31], and Canada [32]. Additionally, one study was developed in multiple countries, including Australia, New Zealand, and Canada [33]. The study designs include cohort studies [24,28–30], case-cohort studies [27], case-control studies [22,23,32,33], prospective control studies [25], and the remaining are considered as translational studies [26,31].

The most frequently used technique was mass spectrometry (MS) and also coupled with liquid chromatography (LC-MS) [24,26] and its derivatives, including liquid chromatography-tandem mass spectrometry (LC-MS/MS) [22,31], ultra-performance liquid chromatography-mass spectrometry (UPLC-MS) [33], ultra performance liquid chromatography – tandem mass spectrometry (UPLC-MS/MS) [27], and liquid chromatography-high-resolution mass spectrometry (LC-HRMS) [23]. Only one study used exclusively MS [32]. Nuclear magnetic resonance (NMR) spectroscopy was also widely employed [24,25,28] along with its derivative, high-resolution magic angle spinning magnetic resonance spectroscopy (HR-MAS MRS) [29,30]. The analysed samples in these studies were serum [22–25,28], plasma [26,27,32,33], urine [28], feces [31] and placental tissue [29,30].

Each metabolomic technique exhibits distinct analytical characteristics, which are summarized in Table 1 and have important implications for metabolite coverage, sensitivity, and confidence of identification.

Mass spectrometry (MS)-based approaches typically rely on a separation step, most commonly liquid chromatography (LC), and provide high sensitivity and broad metabolite coverage. However, they often require extensive sample preparation and can be affected by variable ionization efficiency, matrix effects, and limited reproducibility, which may compromise quantitative accuracy and metabolite identification [34,35]. Targeted LC–MS/MS strategies, including UPLC–MS/MS, improve analytical selectivity and quantitative robustness through multiple reaction monitoring, but require careful method optimization, specialized instrumentation, and often additional confirmatory analyses [36,37]. High-resolution mass spectrometry (LC–HRMS) offers high mass accuracy and broad untargeted coverage, as well as the possibility of retrospective data reprocessing, although it generates large datasets and may still suffer from ambiguity in isomeric metabolite assignment [38]. [39]. In contrast, NMR-based techniques provide inherently quantitative and highly reproducible data, together with direct structural information that facilitates confident metabolite identification. HR-MAS NMR allows the direct analysis of intact tissue, minimizing extraction-related variability, but is limited by lower sensitivity,

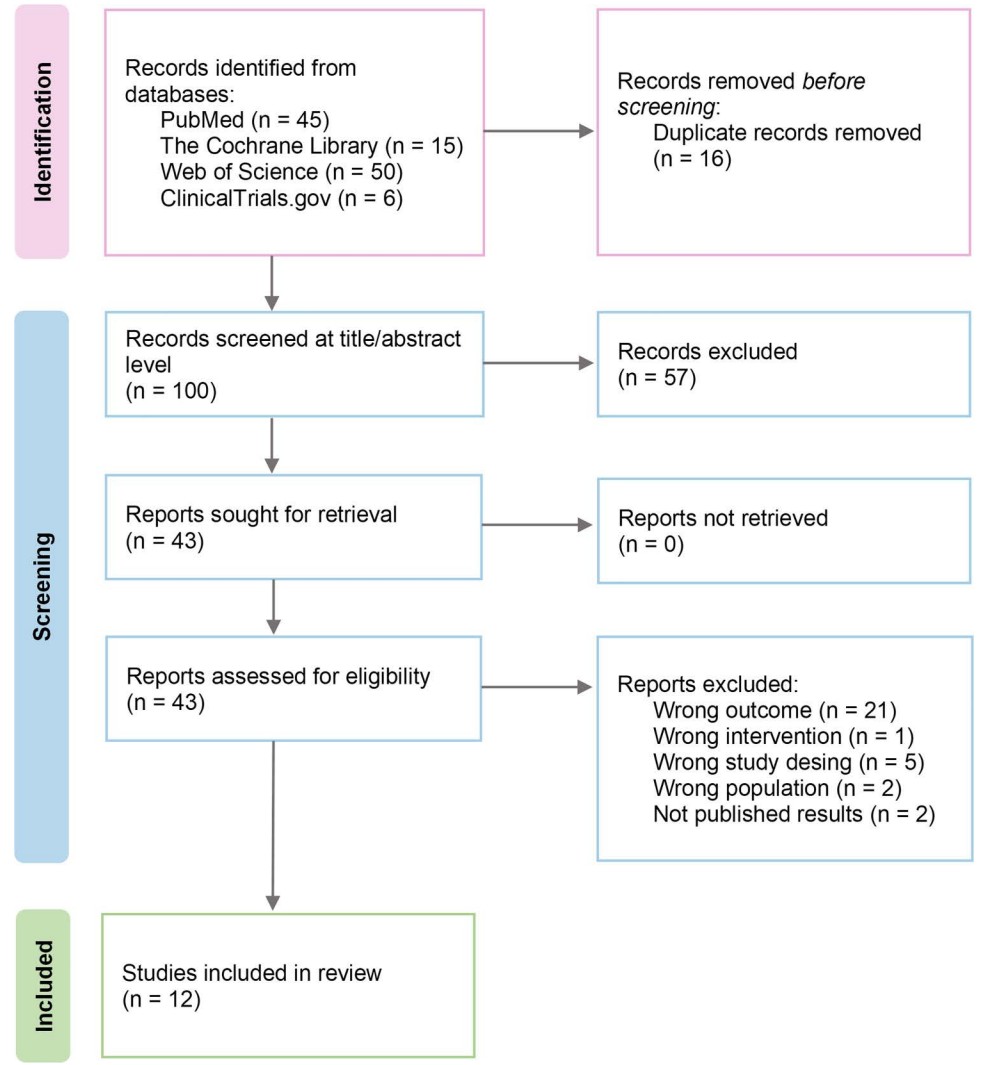

**Fig 1. PRISMA flow diagram summarizing the study selection process.** A total of 112 records were identified through database searching. After removing 16 duplicates, 100 records were screened based on title and abstract, resulting in the exclusion of 57 studies. Of the 43 full-text articles assessed for eligibility, 31 were excluded, and 12 studies met the inclusion criteria for this systematic review.

reduced spectral resolution, and peak overlap [39]. Conventional solution-state NMR shares these advantages and limitations, offering robust quantification and structural insight at the expense of sensitivity compared with MS-based platforms [40].

This methodological heterogeneity across metabolomic technologies has important consequences for the interpretation and comparison of reported metabolite profiles. Differences between studies may reflect not only underlying biological variation but also platform-specific constraints related to metabolite coverage, sensitivity, and confidence of identification. MS-based approaches may detect a wider range of metabolites but are more susceptible to ambiguous annotation of isomeric compounds or analytical artefacts, whereas NMR-based techniques tend to identify fewer, more abundant metabolites with higher structural confidence. Consequently, metabolites consistently detected across multiple studies and analytical platforms are more likely to represent high-confidence biological signals, while technique-specific findings

**Table 1. Comparison of the analytical techniques used in the selected studies.**

| Technique | Advantages | Limitations |
|---|---|---|
| MS | • High sensitivity<br>• Wide metabolite coverage | • Sample preparation/purification<br>• Variable ionization can affect quantification |
| LC-MS | • Improved separation of complex mixtures<br>• Enhanced sensitivity | • Still requires chromatography<br>• Matrix effects<br>• Variable reproducibility |
| LC-MS/MS | • Highly specific targeted quantification<br>• Improved selectivity | • Additional runs for confirmation<br>• Needs two MS instruments |
| UPLC-MS/MS | • Faster analysis<br>• Higher chromatographic resolution | • Requires careful method optimization<br>• Specialized instrumentation |
| LC-HRM | • High mass accuracy<br>• Broad coverage<br>• Retrospective reprocessing of full-scan data | • Large datasets<br>• Advanced data processing required<br>• Lower quantitative specificity |
| HR-MAS MRS | • Direct quantification in intact tissue<br>• No extraction needed<br>• Reduces variability | • Limited sensitivity and spectral resolution<br>• Peak overlap |
| NMR | • Non-destructive<br>• Allows further analyses<br>• Provides quantitative and structural info | • Lower sensitivity compared to MS-based methods |

should be interpreted with caution, as they may partly arise from methodological or analytical limitations rather than true biological differences.

### Participant characteristics

The participants in the selected studies range in age from 18 to 43 years (Table 2). Regarding preeclampsia classification, four studies combined EO-PE and LO-PE cases [22,26,30,33], three focused on LO-PE [25] (with a single EO-PE case included in two of them [23,28]), and five studies did not specify the PE subtype [24,27,29,32]. The majority of the studies were carried out on the first trimester of pregnancy [22–25,27–29], while the others took place in the second [33] and third [26] trimester or after delivery [30,32]. The remaining studies [31] did not specify the gestational age. This variability in case definition and gestational age represents a source of clinical heterogeneity across the included studies. Some of the patients were followed after delivery [24,29,30], or until delivery [22,33], while others were monitored only during the first trimester [23,25,28]. The remaining studies did not report follow-up duration [26,27,31,32].

### 3.1. PE metabolites in serum

The study of *Odibo, A.O. et al.* [22] conducted a nested case-control study that revealed four key metabolites: hydroxy-hexanoylcarnitine, alanine, phenylalanine, and glutamate significantly elevated in maternal blood samples from women who later developed PE compared to pregnant controls (PC; women without HDP) during the first trimester of pregnancy (Table 3). They found that the predictive ability of these metabolites was robust, with an area under the curve (AUC) that ranged from 0.77 to 0.80 when analysed individually. Notably, when these metabolites were combined into a single predictive model, the AUC improved to 0.82 for all cases of PE and even reached 0.85 for EO-PE, indicating a strong predictive capacity. Furthermore, the detection rates of the combined model showed significant practical application, utilizing the combination of significant metabolites, were 50% at a 10% false positive rate (FPR) and 60% at a 20% FPR. For EO-PE, the combined model identified 50% of cases at a 10% FPR, and this detection rate raised to 70% when the FPR was increased to 20%.

**Table 2. Studies and participants characteristics.**

| Authors, year | Country | Study design | Technique | Samples | Maternal age, y (mean±SD or median [range]) | Gestational age (trimester of pregnancy) | Type of PE | Time of follow-up |
|---|---|---|---|---|---|---|---|---|
| Kenny, L.C. et al. 2010 | New Zealand, Australia and Canada | Nested case-control | UPLC-MS | Plasma | Discovery phase: 30.2±4.9 for PE and 30. 4±4.7 for PC Validation phase: 22.0±4.8 for PE and 23.2±5.3 for PC | Second trimester | EO-PE and LO-PE | Whole pregnancy |
| Odibo, A.O. et al. 2011 | United States | Nested case-control | LC-MS/MS | Serum | 32.9±5.5 for PC 30.5±6.2 for PE | First trimester | EO-PE and LO-PE | Until delivery |
| Austdal, M. et al. 2015a | Norway | Cohort | NMR | Serum and urine | 26.0±7.0 for PE 28.0±6.0 for GH 28.0±5.0 for PC | First trimester | LO-PE (1 case of EO-PE) | First trimester |
| Austdal et al. 2015b | Norway | Cohort | HR-MAS MRS | Placental tissue | 25.0±6.0 for PE and FGR 28.0±10.0 for PE severe 34.0±5.0 for PC | First trimes-ter and after delivery | – | After delivery |
| Bahado-Singh, R.O. et al. 2015 | United Kingdom | Prospective control | NMR | Serum | 31.0±7.1 for EO-PE 31.7±5.9 for PC | First trimester | EO-PE | First trimester |
| Austdal, M. et al. 2019 | Norway | Cohort | HR-MAS MRS | Placental tissue | 30 (21–43) for PC 29 (20–40) for PE | After delivery | EO-PE and LO-PE | After delivery |
| Sander, K.N. et al. 2019 | United Kingdom | Translational | LC-MS | Plasma | 28.9±6.7 for PC 30.2±5.1 for PE | Third trimester | EO-PE and LO-PE | – |
| Sovio, U. et al. 2020 | United Kingdom | Case-cohort | UPLC-MS/MS | Plasma | 30 (26–34) for term PE 28 (23–33) for pre-term PE 30 (27–33) for women without pre-term PE | First to third trimester | – | – |
| Ferranti, E.P. et al. 2020 | United States | Nested case-control | HRM and LC-HRMS | Serum | 24.3±4.4 for PC 21.3±2.8 for GH 24.5±4.7 for PE | First trimester | LO-PE (1 case of EO-PE) | 8-14 weeks |
| Harville, E.W. et al. 2021 | United States | Cohort | LC-MS and NMR | Serum | 29.3±5.7 for HDP 29.4±4.8 for GH 29.6±7.4 for PE 30.9±6.9 for PTB 30.3±7.04 for sPTB 28.9±5.01 for PC | First trimester | – | Until 10 weeks after delivery |
| de Almeida, L.G.N. et al. 2022 | Canada | Case-control | MS | Plasma | 35.0±6.5 for NP 32.0±7.1 for PC 35.0±5.9 for GH 32.0±6.1 for PE | Third trimester | – | – |
| Liu, X. et al. 2024 | China | Translational | LC-MS/MS | Feces | 30.14±4.51 for PC 30.48±4.66 for severe PE | – | – | – |

**Abbreviations:** PE, preeclampsia; GH, gestational hypertension; HDP, hypertensive disorders of pregnancy; PTB, preterm birth; sPTB, spontaneous preterm birth; NP, non-pregnant; PC, pregnant controls (women without HDP); EO-PE, early-onset preeclampsia; FGR, fetal growth restriction; HR-MAS MRS, high-resolution magic angle spinning magnetic resonance spectroscopy; NMR, nuclear magnetic resonance; LC-MS/MS, liquid chromatography-tandem mass spectrometry; UPLC-MS/MS, ultra-performance liquid chromatography-tandem mass spectrometry; HRM, high-resolution metabolomics; LC-HRMS, liquid chromatography–high resolution mass spectrometry; y, years; SD, standard deviation; EO-PE, early-onset PE; LO-PE, late-onset PE.

*Austdal, M. et al* [28] utilized Partial Least Squares Discriminant Analysis (PLS-DA) to predict PE, GH, and their combination in pregnant women during the first trimester. The results showed varying prediction accuracy at a 10% FPR: 15% for PE, 33% for GH, and 30% for both combined. Key metabolites linked to these hypertensive disorders included elevated lipid levels, particularly triglycerides, and decreased levels of phosphatidylcholines (especially those associated with

**Table 3. Significant metabolites found in different matrixes.**

| Authors, year | Technique | Samples | Increased metabolites for PE | Decreased metabolites for PE | Statistical/predictive model |
|---|---|---|---|---|---|
| Odibo, A.O. et al. 2011 | LC-MS/MS | Serum | Hydroxyhexanoylcarnitine<br>Phenylalanine<br>Glutamate<br>Alanine | – | Logistic regression modeling |
| Austdal, M. et al. 2015a | NMR | Serum | Triglycerides | Glucose<br>Lactate<br>Alanine<br>Phosphatidylcholine | PLS-DA |
| Bahado-Singh, R.O. et al. 2015 | NMR | Serum | 2-hydroxybutyrate<br>3-hydroxyisovalerate<br>Citrate | Glycerol<br>Acetone | Logistic regression modeling |
| Ferranti, E.P. et al. 2020 | LC-HRMS | Serum | Retinoate*<br>Kynerurine* | sn-glycero-3- Phosphocholine*<br>2′4′-dihydroxyacetophenone* | PLS-DA<br>*These metabolites were found between women who developed PE and those who developed GH and verified by Schymanski Level 1 criteria |
| Harville, E.W. et al. 2021 | LC-MS and NMR | Serum | LC/MS: bolasterone, cerasinone | NMR: asparagine, *N,N*-Dimethylglycine, trimethylamine | Multiple logistic regression |
| Kenny, L.C. et al. 2010 | UPLC-MS | Plasma | Monosaccharide(s)<br>Decanoylcarnitine<br>Oleic acid<br>Docosahexaenoic acidand/or docosatriynoicacid<br>γ-Butyrolactone and/oroxolan-3-one<br>2-Oxovaleric acid and/oroxo-methylbutanoic acid<br>Acetoacetic acid<br>Hexadecenoyl-eicosatetraenoyl-sn-glycerol<br>Di-(octadecadienoyl)-snglycerol<br>Sphingosine 1-phosphate<br>Sphinganine 1-phosphate<br>Vitamin D3 derivatives | 5-Hydroxytryptophan<br>Methylglutaric acid and/oradipic acid | PLS-DA |
| Sander, K.N. et al. 2019 | LC-MS | Plasma | Taurodeoxycholic acid isomer<br>Methionine sulfoxide<br>3-Hydroxyanthranilic acid<br>N1-Methyl-pyridone-carboxamide isomer<br>Urocanic acid<br>Phosphatidylinositol isomer<br>Diacylglycerol isomer | Hydroxyhexacosanoic acid isomer | OPLS-DA |
| Sovio, U. et al. 2020 | UPLC-MS/MS | Plasma | 4-hydroxyglutamate<br>C-glycosyltryptophan | | Forward-stepwise logistic regression |
| de Almeida, L.G.N. et al. 2022 | MS | Plasma | Allatonin<br>Glutamine<br>Histidine | | No predictive model used (ANOVA, Tukey's HSD post-hoc test, hierarchical clustering, metabolic pathway enrichment analysis) |
| Austdal, M. et al. 2015a | NMR | Urine | Creatinine<br>Glycine<br>4-deoxythreonic acid<br>α-hydroxyisobutyrate<br>Histidine<br>Dimethylamine | Hippurate<br>Lactate<br>Proline betaine | PCA |

*(Continued)*

 

**Table 3.** (Continued)

| Authors, year | Technique | Samples | Increased metabolites for PE | Decreased metabolites for PE | Statistical/predictive model |
|---|---|---|---|---|---|
| **Austdal, M. et al. 2015b** | HR-MAS MRS | Placental tissue | Glycerophosphocholine<br>Phosphocholine<br>Aspartate | Ethanolamine<br>Glutamine<br>Glutamate<br>Glycine<br>Taurine<br>Dihydroxyacetone<br>3-hydroxybutyrate<br>Ascorbate | PLS-DA |
| **Liu, X. et al. 2024** | LC-MS/MS | Feces | 5-deoxy-D-glucuronate<br>Phenylpropanoate<br>Agmatine<br>*N*-acetylputrescine | Guanidoacetic acid<br>Valine | PLS-DA |
| **Austdal, M. et al. 2019** | HR-MAS MRS | Placental tissue | Glycerophosphocholine<br>Phosphocholine<br>Aspartate<br>Myoinositol<br>Creatine | Ethanolamine<br>Dihydroxyacetone<br>Taurine<br>Acetate<br>Isoleucine<br>Lysine<br>Leucine<br>Valine<br>Alanine<br>Glucose<br>Threonine<br>Glycine<br>Glutamate<br>Glycerol<br>Dihydroxyacetone<br>3-hydroxybutyrate<br>Ascorbate | No predictive model used (Kruskal–Wallis test, Mann–Whitney U tests (pairwise), with Benjamini–Hochberg FDR correction) |

**Abbreviations:** PE, preeclampsia; GH, gestational hypertension; LC-HRMS, liquid chromatography–high resolution mass spectrometry; LC-MS/MS, liquid chromatography-tandem mass spectrometry; UPLC-MS/MS, ultra-performance liquid chromatography-tandem mass spectrometry; LC-MS, liquid chromatography–mass spectrometry; UPLC-MS, ultra-performance liquid chromatography–mass spectrometry; HR-MAS MRS, high-resolution magic angle spinning magnetic resonance spectroscopy; NMR, nuclear magnetic resonance; MS, mass spectrometry; PCA, principal component analysis; PLS-DA, partial least squares-discriminant analysis; OPLS-DA, orthogonal partial least squares-discriminant analysis; ANOVA, analysis of variance; HSD, honestly significant difference; FDR, false discovery rate.

high-density lipoproteins or HDL). Additionally, lower glucose, lactate and alanine levels were significant in distinguishing hypertensive conditions (Table 3). The relevance of the variability in these metabolites was quantified with variable importance in projection (VIP) scores. Despite the results indicating a correlation between lipid metabolism alterations and hypertension development, the overall prediction accuracy in serum samples was lower compared to the urine analyses also performed in this study. This lower sensitivity was attributed to the challenges in detecting small molecular weight metabolites in serum due to factors like viscosity and signal overlap.

The study conducted by *Bahado-Singh, R.O. et al.* [25] focused on the validation of metabolomic models to predict EO- PE using serum samples collected in the first trimester. Participants were divided into two groups: a discovery group, comprising 30 EO-PE cases and 65 controls, which was utilized for developing the metabolomic biomarker models, and a validation group, containing 20 EO-PE cases and 43 PC, to test the predictive accuracy of the developed algorithms independently. The study employed NMR spectroscopy with an internal spectral database that permitted to analyse around 50 metabolites in each serum sample. Key findings indicated that the metabolite-only model achieved an AUC of 0.835, with a sensitivity of 75% and specificity of 74.4%. In this model, the significant biomarkers predictors were 2-hydroxybutyrate, 3-hydroxyisovalerate, acetone, citrate and glycerol (Table 3). When combined with UtAPI, the performance improved

significantly, resulting in an AUC of 0.916, with sensitivity and specificity rates increasing to 90% and 88.4%, respectively. The validation of the models showed reproducibility, as the diagnostic accuracy in the validation group mirrored that of the discovery group, confirming the robustness of the algorithms developed.

*Ferranti, E.P. et al.* [23] identified specific metabolites in serum samples obtained during the first trimester as significantly differentially regulated between women who developed PE and those who developed GH. A total of 470 significant features were identified between PE and PC, and 388 were identified between GH and controls. A PLS-DA identified 169 significant features based on both *p*-value < 0.05, using generalized linear regression methods, and a VIP score > 2, between PE and GH. Only metabolites verified by Schymanski Level 1 criteria showing significant differences between PE and GH were reported in detail. They found increased levels of retinoate and kynurenine, and decreased levels of sn-glycerol-3-phosphocholine and 2'4'-dihydroxyacetophenone in PE compared to GH (Table 3). In addition to these metabolites, they also identified significantly enriched metabolic pathways compared to PC. These pathways included alterations in porphyrin metabolism, steroid hormone biosynthesis, retinol metabolism, and arachidonic acid metabolism in PE, as well as changes in fructose and mannose metabolism and various amino acid metabolism pathways in GH.

Five groups of populations were analysed by *Harville, E.W. et al.* [24] related to pregnancy complications in the first trimester: women with PE, GH, HDP and preterm birth (PTB) or spontaneous preterm birth (sPTB). Advanced techniques such as UPLC-HR-MS and NMR were used to identify metabolites from serum samples associated with these disorders. In the case of PE, several predictive metabolites, such as bolasterone, cerasinone and unidentified signals, significantly improved the model's predictive ability with an AUC of 0.95. Additionally, the NMR analysis revealed that asparagine, *N,N*-dimethylglycine and trimethylamine were linked to lower probability of PE, with an AUC of 0.177. For GH, predictive signals were found using UPLC-HR-MS, such as 2,6-di-tert-butyl-4-hydroxymethylphenol, which showed an odds ratio (OR) of 2.57, with an AUC of 0.945, significantly improving the model. However, no predictive metabolites were identified with NMR. In HDP, which include both PE and GH, UPLC-HR-MS identified predictive signals such as glycochenodeoxycholic acid 3-glucuronide, which showed an OR of 5.43, and unidentified signals associated with lower odds of HDP, with an AUC of 0.954. NMR also contributed with signals associated with lower odds of HDP, such as the combination of asparagine/albumin, with an AUC of 0.0412. Finally, for PTB and sPTB, predictive metabolites were identified using UPLC-HR-MS, such as {3-[(2E)-3-phenylprop-2-enoyl]phenyl} oxidanesulfonic acid in PTB and anisoxide in sPTB, among others, which improved the AUC of the models, reaching 0.821 for PTB and 0.865 for sPTB (Table 3). NMR found metabolites like threonine and urea, associated with a reduced risk of PTB and sPTB, though they did not significantly contribute to the predictive value of the model.

## 3.2. PE metabolites in plasma

*Kenny, L.C. et al*. [33] conducted a two-phase metabolic profiling study using UPLC-MS to identify plasma metabolic biomarkers for predicting PE in the second trimester of pregnancy. In the first phase, the discovery phase, they identified 14 metabolites that were discriminant for PE, including fatty acids, acylcarnitines, and amino acid–related compounds, among others (Table 3). The multivariate model developed to predict these metabolites showed an odds ratio of 36 (95% Confidence Interval, CI: 12–108) for developing PE, with an AUC of 0.94. These findings were further validated in an independent case-control study, where the same 14 metabolites produced an odds ratio of 23 (95% CI: 7–73) and an AUC of 0.92. They performed a PLS-DA model using the data from both studies demonstrating that a combination of 14 metabolites that represents systemic interactions in the metabolome is sufficient to generate a robust predictive model with an AUC greater than 0.9. This model showed promising detection performance, with subsequent PE detection rates of 77% and 73% in the discovery and validation datasets at a 10% false-positive rate, and false-positive rates of 21% and 24% at a 90% detection rate.

The separation of metabolites into polar and apolar fractions allowed *Sander, K.N. et al.* [26] to identify 35 metabolites associated with PE at third trimester of pregnancy, using LC-MS, although not all could be fully characterized. A total

of 4379 peaks corresponding to polar compounds and 3153 peaks for apolar compounds were initially detected. From the full dataset, 37 metabolites were selected through both univariate and multivariate analysis. After excluding two compounds identified as drugs, 35 remained as potential biomarkers. These showed clear group separation in Principal Component Analysis (PCA) (and OPLS-DA, with a highly significant model ($R^2X(cum)$ = 0.629, $Q^2(cum)$ = 0.648, $p = 5.5 \times 10^{-12}$), achieving 91% prediction accuracy and 100% specificity and an AUC value of 0.964. Among them, nine compounds were tentatively identified using metabolomics databases. In the polar fraction, taurodeoxycholic acid isomer, methionine sulfoxide, 3-hydroxyanthranilic acid, N1-methyl-pyridone-carboxamide isomer, and urocanic acid were identified. The apolar fraction contained hydroxyhexacosanoic acid isomer, two types of diacylglycerols, and glycerophosphoinositol (Table 3).

Sovio, U. et al. [27] identified 4-hydroxyglutamate as a novel biochemical predictor of PE, demonstrating a significantly stronger association compared to existing biomarkers like PlGF and PAPP-A. They analised serum samples of women with PE (delivering at term and pre-term) and women without pre-term PE with 12, 20, 28 and 36 weeks of gestational age (wkGA). Statistical analyses revealed that among the 1193 untargeted metabolites screened, 4-hydroxyglutamate showed promising predictive capabilities (Table 3). Specifically, the predictive ability of 4-hydroxyglutamate at 12 wkGA had an AUC of 0.673. At 20 wkGA, it remained strong, and by 28 wkGA, its predictive power was further enhanced, resulting in an AUC of 0.848 when combined with C-glycosyltryptophan. Although C-glycosyltryptophan also demonstrated some potential as a predictor, its strong association with pre-term PE was primarily evident at 28 weeks, with less consistent results at earlier gestation periods. The addition of C-glycosyltryptophan to predictive models improved their overall capability, but its clinical utility is considered limited compared to that of 4-hydroxyglutamate. The predictive accuracy of these metabolites was validated in an external cohort of samples from Born, in Bradford, highlighting their robustness across diverse populations. This evidence suggested that 4-hydroxyglutamate could serve as a valuable biomarker for early identification of women at high risk for PE.

The metabolomics analysis conducted in the study of de Almeida, L.G.N. et al. [32] highlighted significant alterations in the metabolic profiles of plasma samples from participants in the third trimester of pregnancy in various groups: non-pregnant (NP), PC, GH and PE. A total of 52 metabolites were identified during the analysis. Among the metabolites examined, three specific metabolites (allantonin, glutamine and histidine) exhibited markedly elevated levels in the PE group compared to PC (Table 3). Specifically, allantoin showed significant increase, with higher intensities in PE ($p < 0.01$) when compared to PC, while glutamine and histidine also demonstrated significant increases in PE ($p < 0.05$). The distinct metabolic profiles associated with PE were further supported by hierarchical clustering analysis, which effectively separated the NP group from the other three groups (PC, GH, and PE).

### 3.3. PE metabolites in urine

Austdal, M. et al. [28] demonstrated that first-trimester urinary metabolomic profiles had the potential to predict both PE and GH. Specifically, multivariate analyses revealed that at a 10% false positive rate, PE could be predicted with a sensitivity of 51.3%, while GH was predicted with a sensitivity of 40%, and both conditions combined at 37%. Hippurate was found to be significantly decreased in women who subsequently developed either PE or GH. In women who later developed PE, the urinary levels of creatinine, glycine, 4-deoxythreonic acid, $\alpha$-hydroxyisobutyrate, histidine and dimethylamine were increased, while hippurate, lactate, and proline betaine were decreased (Table 3). For those who developed GH, there was an additional decrease in urinary citrate excretion alongside the changes observed in PE. The PCA of the $^1H$ NMR spectra of urine samples showed a distinct clustering of women who were set to develop PE or GH, indicating a clear difference in their urinary metabolic profiles compared to PC. Notably, combining the urinary hippurate/creatinine ratio with maternal mean arterial pressure (MAP) and a variable for maternal age improved the prediction rates for PE in a logistic regression model (AUC of 0.778) compared to using UtAPI combined with MAP and age (AUC of 0.738).

### 3.4. PE metabolites in feces

*Liu, X. et al.* [31] revealed a significant alteration in the fecal metabolome in women with severe PE compared to PC. Through untargeted metabolomic analysis using LC-MS/MS, a total of 686 fecal metabolites were identified after normalization. Further analysis pointed 31 metabolite features exhibiting distinct enrichment patterns between the two groups. Notably, guanidoacetic acid and valine were found to be significantly more abundant in PC, while 5-deoxy-D-glucuronate, phenylpropanoate, agmatine, and N-acetylputrescine were enriched in the severe PE group (Table 3). KEGG pathway analysis indicated that metabolites with higher concentrations in PC were significantly enriched in arginine and proline metabolism and glycine, serine and threonine metabolism. Conversely, the inositol phosphate metabolism and pentose and glucuronate interconversions pathways were enriched by metabolites more abundant in severe PE. Furthermore, the study identified *Limosilactobacillus fermentum* as a key bacterial biomarker associated with severe PE. Notably, its abundance showed a significant negative correlation with the metabolites phenylpropanoate ($p < 0.001$) and agmatine ($p < 0.05$), suggesting a potential metabolic interaction. In addition, phenyl-propanoate exhibited excellent diagnostic performance, with an AUC of 98.57%, in distinguishing women with severe PE from PC. Co-occurrence network analysis between gut bacteria and fecal metabolites showed strong inter-relationships, with metabolites more abundant in PC generally clustering separately from those elevated in severe PE. *Limosilactobacillus fermentum* showed a significant positive correlation with bacterial species and metabolites enriched in PC.

### 3.5. PE metabolites in placental tissue

The study of *Austdal, M. et al.*[29] identified 25 metabolites in a HR-MAS MRS analysis of placental biopsies. A univariate analysis revealed that 12 metabolites showed significant differences between placentas from pregnancies with PE and PC after correction for multiple testing. Specifically, in preeclamptic placentas, a decrease in the levels of ethanolamine, glutamine, glutamate, glycine, taurine, dihydroxyacetone, 3-hydroxybutyrate and ascorbate was observed, while the levels of glycerophosphocholine were found to be increased. A decrease in valine, threonine, and lysine was also observed in preeclamptic placentas. PCA indicated that an increase in aspartate, phosphocholine, and glycerophosphocholine, and a decrease in glutamate, taurine, ascorbate and glutamine corresponded with the preeclamptic phenotype. PLS-DA models showed that the levels of glycerophosphocholine, phosphocholine, and aspartate were increased, while the levels of ethanolamine, taurine, glutamate, ascorbate, and glycine were decreased in PE compared to PC (classification accuracy 92.7%, $p < 0.001$). In severe PE, an increase in choline, lysine, alanine, glucose, and myo-inositol, along with a decrease in 3-hydroxybutyrate, was observed compared to non-severe PE (accuracy 83.2%, $p = 0.003$) (Table 3).

In 2019, *Austdal, M. et al.* [30] studied three distinct metabolic placental groups through hierarchical clustering of metabolic profiles obtained by HR-MAS MRS: the normal placenta group, the moderate placental dysfunction group, which contained almost all LO-PE pregnancies and a substantial number of PC; and the severe placental dysfunction group, that included more women with EO-PE, fetal growth restriction (FGR) and preterm delivery, and lower placenta weight. The normal placenta group was metabolically distinct, showing significant differences in 21 metabolites involved in pathways like BCAAs degradation, glycine, serine, and threonine metabolism, and phospholipid biosynthesis. Compared to the normal placenta group, in the moderate placental dysfunction group, the levels of glycerophosphocholine, phosphocholine, aspartate, and myoinositol were significantly increased, while ethanolamine, dihydroxyacetone, taurine, acetate, isoleucine, lysine, leucine, valine, alanine, glucose, threonine, glycine, and glutamate were significantly decreased. In the severe placental dysfunction group, the levels of phosphocholine, aspartate, and creatine were significantly increased, while ethanolamine, glycerol, dihydroxyacetone, taurine, acetate, lysine, valine, glucose, 3-hydroxybutyrate, ascorbate, threonine, glycine, and glutamate were significantly decreased (Table 3).

## 4. Discussion

Although several studies have applied metabolomics to identify biomarkers for PE across multiple biological matrices (serum, plasma, urine, feces, and placental tissue), overlap in individual metabolites remains limited. Importantly, interpretation requires explicit consideration of both gestational timing and biological compartment. First-trimester maternal samples primarily provide early predictive or prodromic signals, whereas placental tissue analysed at delivery reflects late descriptive changes associated with established disease. Likewise, maternal biofluids capture systemic maternal physiology, while placental profiles represent local tissue metabolism. Accordingly, apparent inconsistencies in the direction of metabolite changes should be interpreted as phase- and compartment-dependent signatures rather than purely analytical discrepancies. In addition, EO-PE and late-onset LO-PE are increasingly recognized as partially distinct clinical entities with different dominant pathophysiological drivers. EO-PE is mainly associated with placental dysfunction and impaired placentation, whereas LO-PE appears more closely linked to maternal cardiovascular and metabolic maladaptation with relatively preserved placental function [41]. These distinctions likely contribute to the variability observed across metabolomic studies.

During the first trimester of pregnancy, alanine was identified as a discriminant metabolite in two studies that used serum (Fig 2). However, the findings regarding this metabolite were inconsistent, one study that empoloyed LC-MS/MS reported elevated levels in preeclamptic patients [22], while the other observed decreased concentrations through NMR [28]. Alanine was also identified as a decreased biomarker for PE in placental tissue using HR-MAS MRS [29]. Rather than being interpreted primarily as analytical discrepancies, these differences may reflect compartment- and stage-dependent metabolic processes. First-trimester maternal serum profiles likely capture early systemic metabolic adaptation, whereas placental tissue analysed at delivery reflects established local dysfunction.

During pregnancy, the metabolic demands of the mother and the developing fetus are high. In healthy pregnancies, increased maternal glucose production, altered amino acid metabolism, and adaptive changes in placental blood flow support fetal growth [42].

Within this physiological context, alanine plays a central role in maternal–fetal nitrogen transport and gluconeogenic flux through the glucose–alanine cycle. Early increases in circulating alanine may therefore reflect enhanced systemic amino-acid mobilization or altered hepatic substrate handling during prodromic stages of disease.

Alterations in alanine metabolism have been reported in PE, with elevated alanine levels particularly observed in severe cases. These changes may reflect impaired transamination and hepatic dysfunction, as suggested by increased serum alanine aminotransferase (ALT) levels [43,44]. Conversely, reduced alanine levels observed in placental tissue may indicate impaired local amino-acid turnover, mitochondrial dysfunction, or altered transamination capacity within the placenta once the disease is established. This supports the hypothesis that amino acid processing capacity is compromised in PE, likely due to oxidative stress and inflammation [45].

Lactate was reported in one study as a decreased common metabolite between serum and urine through NMR in the first trimester of pregnancy [28]. Although this observation comes from only one study, it may reflect some pathophysiological changes in PE.

Lactate has been previously associated with severe PE and adverse outcomes, including liver and kidney disfunction [46]. However, the decrease observed in first-trimester maternal samples likely reflects early systemic metabolic regulation rather than placental hypoxia. At this gestational stage, placental dysfunction is not yet fully established, and mechanisms characteristic of advanced disease cannot be directly extrapolated. Alanine and lactate share a common metabolic precursor, pyruvate, which serves as a key regulatory node. Lower circulating lactate in early pregnancy may indicate increased oxidative utilization of pyruvate, enhanced hepatic gluconeogenesis, or shifts in maternal substrate allocation associated with early cardiometabolic stress. Such systemic adaptations may precede the onset of overt placental pathology and therefore represent prodromic metabolic signatures rather than consequences of established hypoxia. Under hypoxic conditions such as those seen in preeclamptic placentas, pyruvate is preferentially converted to lactate via lactate dehydrogenase (LDH) through the Cori Cycle,

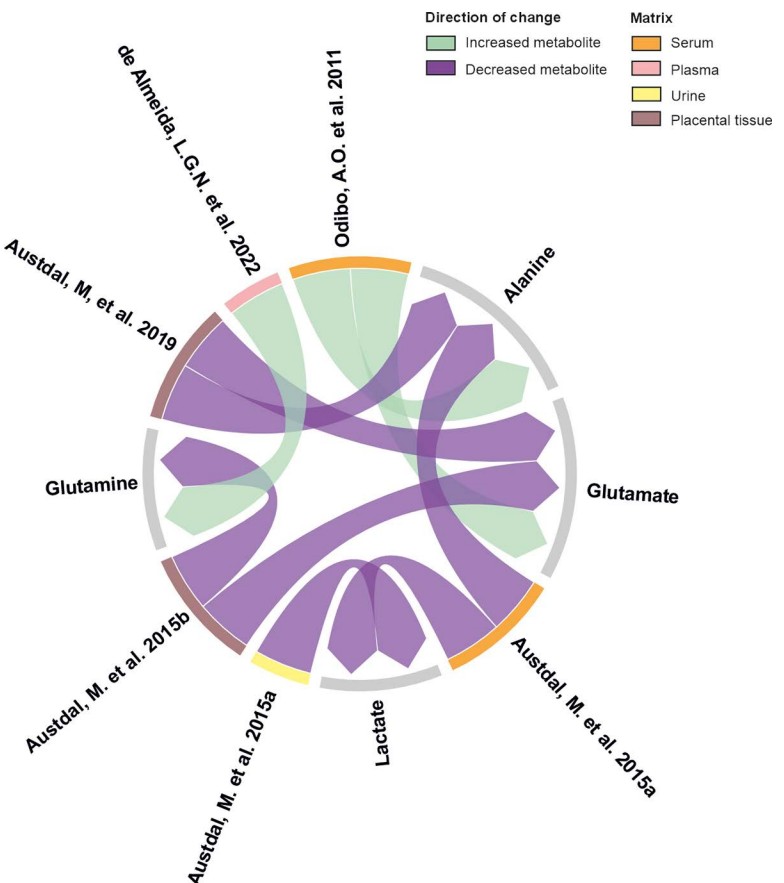

**Fig 2. Chord diagram depicting shared biomarkers among selected studies.** This figure illustrates the relationships between studies included in the systematic review and the metabolites identified in common across them. Only those studies that share at least one biomarker are represented. The diagram was generated using RawGraphs.io.

while simultaneously it can be transaminated to alanine by ALT [47,48]. Increased ALT activity may reflect both hepatic stress and an adaptive attempt to sustain gluconeogenesis via the glucose–alanine cycle [44,49]. This cycle facilitates nitrogen transport and offers an alternative route for pyruvate utilization, thus linking alanine and lactate metabolism tightly (Fig 3A) [50].

Importantly, this hypoxia-driven metabolic routing is more characteristic of later stages of established disease rather than the early gestational window evaluated in most first-trimester maternal metabolomic studies. In this early context, the simultaneous decrease in lactate and alanine reported across several cohorts may instead reflect increased hepatic utilization of both metabolites as gluconeogenic substrates, enhanced mitochondrial oxidative use of pyruvate, or metabolic heterogeneity across PE subtypes, particularly in late-onset forms. These alternative interpretations help reconcile apparently divergent findings within a stage-dependent metabolic framework.

Other common identified metabolites in several studies include glutamate, which was found at higher levels in serum by LC-MS/MS in the first trimester [22] but at lower levels in placental tissue using HR-MAS MRS [29,30], and glutamine, which showed increased concentrations in plasma through MS in the second trimester [32] and decreased concentrations in placental tissue employing HR-MAS MRS [29] (Fig 2). These direction-dependent differences are therefore better interpreted within the previously described compartment- and stage-specific metabolic framework rather than as contradictory findings.

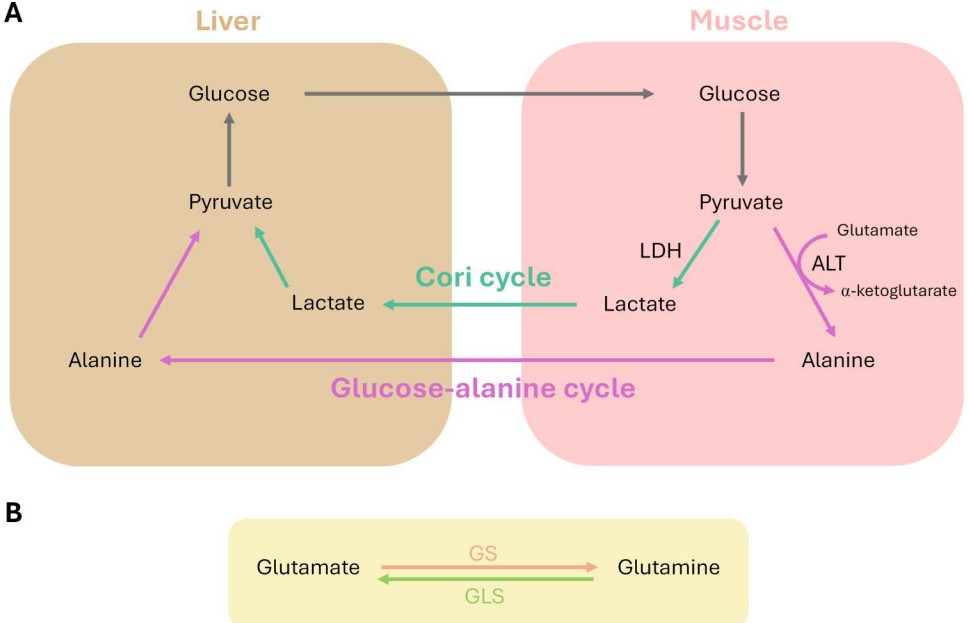

**Fig 3. Key metabolic pathways linking lactate, alanine, glutamate, and glutamine. (A)** Diagram illustrating the interplay between the Cori cycle (green arrows) and the glucose–alanine cycle (pink arrows), connecting liver and muscle metabolism. Pyruvate serves as a central intermediate, converted to lactate via lactate dehydrogenase (LDH) and to alanine via alanine aminotransferase (ALT), using glutamate as a nitrogen donor. **(B)** Interconversion between glutamate and glutamine. Glutamate is converted to glutamine by glutamine synthetase (GS), while glutamine is hydrolyzed back to glutamate by glutaminase (GLS).

The observation that glutamate and glutamine appear in both compartments suggests that they participate in shared metabolic axes linking maternal metabolism and placental function, but with different directional responses depending on stage and tissue context. Glutamine serves as a precursor for glutathione, one of the body's most important endogenous antioxidants [51]. In PE, reduced glutathione levels have been observed [52], suggesting a potential disruption in glutamine metabolism. Altered maternal glutamine availability may therefore reflect early systemic oxidative stress and immune activation, processes known to precede clinical manifestations of PE.

Glutamine also plays a role in immune regulation and has been shown to modulate systemic inflammation, a hallmark of PE [53]. Glutamate is the principal excitatory neurotransmitter in the central nervous system, and its excessive accumulation can lead to excitotoxicity, a pathological process associated with neuronal injury or cellular death [54]. Beyond its neural role, glutamate is a central node in nitrogen transfer and amino acid interconversion, and elevated maternal levels may reflect altered amino acid flux or inflammatory activation rather than direct placental dysfunction. Notably, it has been shown that glutamate can increase the risk of PE by 5.5 times [55].

Glutamate is also involved in the glucose-alanine cycle by serving as the main source of nitrogen for the synthesis of alanine by ALT (Fig 3A) and it also acts as a substrate for glutamine synthesis by glutamine synthetase (GS) [56]. Glutamine can also be converted to glutamate by the action of the glutaminase (GLS) (Fig 3B) [57]. Together, these interconversions suggest that perturbations in the glutamate–glutamine axis may reflect systemic metabolic stress early in pregnancy and local metabolic exhaustion in placental tissue at later stages, providing a physiologically coherent explanation for the differing directions reported across studies. Within this broader framework, glutamate and glutamine metabolism can also be interpreted as functionally integrated into the pyruvate–alanine–lactate metabolic axis, as transamination reactions couple amino acid metabolism to central carbon flux. This network-based perspective unifies

previously discussed pathways and provides a more physiologically coherent interpretation of the coordinated metabolic alterations reported across studies.

Although only a few metabolites are consistently identified across studies, metabolomics remains a highly valuable and innovative approach for uncovering novel biomarkers in PE. This technique offers unparalleled sensitivity and the unique ability to capture real-time changes in the biochemical phenotype, reflecting the dynamic physiological and pathological processes underlying the disease. Metabolomics directly measures the end products of cellular metabolism, providing a closer snapshot of functional alterations [58,59].

Therefore, the discrepancies in observed levels may reflect differences in the maternal-fetal metabolic environment, such as hypoxia, inflammation, and hepatic dysfunction. Notably, the intrinsic metabolic plasticity of pregnancy, driven by intense systemic adaptations, may amplify the metabolomic signals associated with pathological deviations such as PE. This biological context could be leveraged to improve early biomarker discovery, particularly in longitudinal studies aimed at translating these findings into clinically actionable and digitally integrated biomarkers.

Metabolomics also presents limitations that must be acknowledged. It is highly sensitive to external variables such as diet, microbiota, or medication, which poses a major challenge for reproducibility and clinical validation of findings [60]. Moreover, the integration of metabolomics with clinical data and other molecular techniques holds great promise for developing comprehensive predictive models and personalized medicine strategies in PE. However, this review is subject to certain methodological limitations. The relatively small number of included studies, combined with substantial heterogeneity across them, precluded data pooling and limited the feasibility of more robust quantitative analyses. Methodological and data preprocessing differences across studies may further contribute to variability in reported metabolomic findings. Studies differed widely in normalization strategies, scaling methods, quantification approaches and data processing workflows. Such differences can affect metabolite detection and relative abundance estimates, potentially leading to apparent discrepancies in the metabolites identified across studies. Recognizing these methodological challenges highlights the importance of standardized workflows but does not diminish the potential of metabolomics to uncover meaningful biological insights.

From a clinical perspective, metabolomics using MS and NMR holds significant potential for early risk assessment, disease stratification, and monitoring of PE. MS-based platforms offer high sensitivity and broad metabolite coverage, making them particularly suited for exploratory biomarker discovery, while NMR-based approaches provide highly reproducible and quantitative data with minimal sample preparation, facilitating longitudinal studies and comparisons across cohorts. In other clinical areas, metabolomics has shown promise for early detection and risk prediction, identifying pre-diagnostic serum metabolites associated with breast cancer risk [61] or serum biomarkers of disease progression in multiple sclerosis [62]. Nevertheless, the direct translation of metabolomic findings into clinical practice for PE is currently limited by heterogeneity in study designs, including differences in gestational age at sample collection, PE subtype, cohort characteristics, and analytical workflows. Given this variability, future research should prioritize more targeted and standardized study designs. Focusing separately on EO-PE and LO-PE, harmonizing sample collection protocols and adopting reproducible analytical methodologies would likely improve the robustness and clinical relevance of identified biomarkers. Metabolomics, at present, should be viewed as a complementary research tool rather than a standalone diagnostic approach.

In summary, despite the limited overlap of individual metabolites across studies, the available evidence indicates that early-pregnancy metabolomic profiling can capture biologically meaningful signals associated with later PE development. However, the small number of studies, differences in gestational timing, PE subtype definitions, analytical platforms, and the inconsistent directionality observed for some metabolites limit the strength of current inferences. Rather than providing definitive predictive markers at present, the existing literature is better interpreted as identifying plausible metabolic pathways and interconnected physiological axes that merit further investigation. By moving beyond the compilation of individual findings, this integrative synthesis highlights how apparently disparate metabolite alterations

converge into coordinated metabolic networks, offering a more coherent physiological framework for understanding early PE pathophysiology.

## Supporting information

**S1 Appendix. Review Objectives and Research Questions.**
(DOCX)

**S2 Appendix. PICOS Criteria for Study Selection.**
(DOCX)

**S1 Table. Critical appraisal for the selected studies.**
(DOCX)

**S1 Fig. Risk of bias assessment of the included studies.** Risk of bias assessment across the included studies using the CADIMA criteria. The figure summarizes the evaluation of each study based on four domains: D1 (sample size sufficiency for metabolomic profiling), D2 (clarity of inclusion/exclusion criteria), D3 (biomarker measurement), and D4 (external validity). Judgements are categorized as low risk (+), unclear risk (–), or high risk (x). Overall risk of bias reflects the combined judgement across all four domains. The visualization was generated using the robvis tool [63].
(DOCX)

**S2 Table. Excluded studies at title/abstract screening.**
(DOCX)

**S3 Table. Excluded studies at fulltext screening.**
(DOCX)

**S3 Appendix. PRISMA Checklist for systematic reviews.**
(DOCX)

## Author contributions

**Conceptualization:** Maria Dolores Jara Montes, Ana Cristina Abreu, Manuel A. Rodríguez Maresca, Ana María Fernández Alonso.

**Data curation:** Celia García-Mañas.

**Formal analysis:** Celia García-Mañas.

**Funding acquisition:** Ignacio Fernandez.

**Investigation:** Celia García-Mañas, Maria Dolores Jara Montes, Ana Cristina Abreu.

**Methodology:** Celia García-Mañas, Maria Dolores Jara Montes, Ana Cristina Abreu, Ana María Fernández Alonso.

**Project administration:** Ana María Fernández Alonso.

**Supervision:** Ana Cristina Abreu, Ignacio Fernandez.

**Validation:** Ana Cristina Abreu, Manuel A. Rodríguez Maresca, Ignacio Fernandez, Ana María Fernández Alonso.

**Visualization:** Ignacio Fernandez.

**Writing – original draft:** Celia García-Mañas.

**Writing – review & editing:** Ana Cristina Abreu, Manuel A. Rodríguez Maresca, Ignacio Fernandez, Ana María Fernández Alonso.

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
