## [Decision Letter · Decision Letter 0]

10 Dec 2025

PONE-D-25-46421Untargeted Metabolomics for the Early Detection of Preeclampsia: A Systematic Review of Human StudiesPLOS One

Dear Dr. Fernandez,

Thank you for submitting your manuscript to PLOS ONE. After careful consideration, we feel that it has merit but does not fully meet PLOS ONE’s publication criteria as it currently stands. Therefore, we invite you to submit a revised version of the manuscript that addresses the points raised during the review process.

We look forward to receiving your revised manuscript.

Kind regards,

María Teresa Llinás

Academic Editor

PLOS One

**Journal Requirements:**

3.. Thank you for stating the following financial disclosure:

“C. G-M. thanks Agencia Española Contra el Cáncer (AECC) for a predoctoral grant (PRDAM234245GARC).”

**Additional Editor Comments:**

Thank you for your efforts in preparing a comprehensive review of untargeted metabolomics applied to preeclampsia. The topic is timely, scientifically relevant, and of clear potential impact for advancing early detection strategies. However, there are important aspects that need to be clarified.

The reviewers’ detailed comments are enclosed and should guide a thorough revision of your manuscript. We encourage you to restructure the results and discussion to improve interpretability, integrate a clearer evaluation of the clinical utility of metabolomics (including the directionality of metabolite changes and matrix-specific findings), and provide a more robust methodological critique. Addressing these issues will substantially strengthen the manuscript and enhance its value to the field.Addressing these issues will substantially strengthen the manuscript and enhance its value to the field.

Reviewers' comments:

Reviewer's Responses to Questions

**Comments to the Author**

1. Is the manuscript technically sound, and do the data support the conclusions?

Reviewer #1: Yes

Reviewer #2: Partly

2. Has the statistical analysis been performed appropriately and rigorously? 

Reviewer #1: N/A

Reviewer #2: N/A

3. Have the authors made all data underlying the findings in their manuscript fully available?

Reviewer #1: Yes

Reviewer #2: Yes

4. Is the manuscript presented in an intelligible fashion and written in standard English?

Reviewer #1: Yes

Reviewer #2: Yes

5. Review Comments to the Author

Reviewer #1: Dear Authors,

Thank you for providing a well planned, rigorous and inclusive review. The review summarises existing literature on available studies that use metabolomics in MS or NMR methods to investigate differences between women with preeclampsia and healthy pregnant controls.

I will start with some overarching questions and suggestions for improvement, then go on to details.

Although the authors state that a metaanalysis is not possible due to differing methods between the studies, an extended analysis on the clinical utility of metabolomics in preeclampsia is warranted, especially limited to MS or NMR metabolomics. Figure 2 attempts to summarise findings on the same metabolites between studies, but ignore differences in direction of change between studies. Would it be possible to include the direction of change, maybe as a color code? The figure also conflates the different study matrices like placenta and serum/plasma. It would be nice to see in the figure which matrix is discussed when referring to a metabolite.

Similarly, when describing each study, a note of timing of pregnancy would benefit interpretability.

Authors cite differing metabolic states of patients as a reason for the differences between studies, but do not discuss methodological and preprocessing differences between studies, for example normalization strategies and quantification methods. The review would benefit from a short discussion on these limitations as well.

Given the heterogeneity of findings in the studies, what is a way forward for the field. Validation of previous findings, larger exploratory studies, more specific and reproducible methodology? please discuss.

Line 57, EO and LO PE are described as “usually occurring”, “normally presents” before/after 34 weeks. This is the actual definition of EO/LO PE so there is no usuall/normally, these always occur in their defined GA periods.

Below, some more on grammar/writing details, which will help the manuscript become more readable and concise:

• Authors appear to follow abbreviation standards from the included papers, which is confusing, and do not always abbreviate at all, or define abbreviations that are not used again. Examples:

• Preeclampsia is written in full several times even though an abbreviation is provided at the start. Examples: PE defined line 48, preeclampsia spelled out line line 75, 102, 222,340 etc

• Weeks of gestation defined on line 340 as wkGA but then not used again

• Gestational hypertension defined as GHT on line 258 (not actually defined in the paper) and as GH on line 267

• Pregnant controls are inconsistently defined as PC (357), “Healthy controls”(394), “normotensive pregnancies”, “normal pregnancies”, “healthy women”. Although different nomenclature might reflect different populations, the review would improve readability with consistent naming of non-pregnant women, pregnant women without HDP, and PE/GH/HDP populations.

• PLSDA defined several times: (221,255, 293). PCA mentioned at line 377, but not at first mention (328)

• Some metabolites are inconsistently named: L-glutamine, L-histidine (362, 359), L-valine (390) valine (439) glutamine, histidine (374). Similarly, please refer to metabolites by a consistent name: citrate (367) is the same metabolite as ascorbate (416). Accordingly, does this change your analysis if you did not realize they were the same?

• Authors refer to results from logistic regression studies both as AUC (area under curve) (213, 243, 272)and ROC curve area (315, 330), please stay consistent.

• FGR not defined (line 432)

• Authors suggest that differences in metabolites between studies might be due to metabolic profiles of the patients (458), this is a circular argument: differences in metabolites are due to differences in metabolites. Please refine your argument.

• Wording on line 70 is confusing, please rephrase “PlGF just can predict less 90% of early PE cases”. Also do you mean EO-PE as defined earlier?

• Paragraph on line 84 is not very relevant, describing cancer and Covid-19. It can be shortened to a few sentences. Perhaps a better use of this space is to provide an overview of other more or less validated biomarkers of preeclampsia, for example sFLT, PlGF and PAPP-A? This would help when these are discussed later in line 338.

• Table 1, column “Age, average years” is unclear. What do the two numbers mean in row 1,3,6 etc. Only last row has a measure of confidence, are these standard deviance?

• Line 221, extra period after citation

• Uterine doppler PI mentioned line 246 but defined with abbreviation at line 383

• Line 273 refers to an AUC of 0, is this correct?

• Throughout the description of the included studies, a lot of metabolites are listed. These are repeated in the table 2. Would it be possible to reduce the mentioned metabolites in text to a few most interesting ones, to improve readability? For example lines 303-310.

• Table 2, abbreviations not defined in table heading/footer.

• Seems unnecessary to define abbreviations that are not used again, lines 296-299.

• Line 408, does this refer to healthy pregnant women or non pregnant women?

• Line 430, unclear grammar. Should it be “contains almost all”?

Reviewer #2: This systematic review provides an updated synthesis of untargeted metabolomics studies applied to the early detection of preeclampsia. Following a structured PRISMA based search process, the authors identified 13 studies covering different designs (cohort, case–control, case-cohort, validation, and translational studies) and a wide range of biological matrices, including serum, plasma, urine, feces, and placental tissue. The review highlights several recurrent metabolites, particularly alanine, lactate, glutamate, and glutamine, inked to physiopathological processes relevant to preeclampsia. The authors conclude that metabolomics holds strong potential for identifying sensitive biomarkers capable of predicting or diagnosing the disease at an early stage.

Overall, the topic is scientifically compelling and of considerable interest, given the need for improved early detection strategies in preeclampsia. However, there are important methodological limitations that affect the strength of the conclusions and reduce the validity of the review.

Major

1.- The PICO question is not fully adhered to in the final study selection. The article by Liu et al. (reference 30) does not compare the group of women with preeclampsia to women with healthy pregnancies. The authors should either exclude it from the review or justify its inclusion in the text and explain the implications of this choice.

2.-The authors do not conduct any subgroup analyses despite the evident heterogeneity among the included studies. Instead, they restrict their approach to presenting individual descriptions of each study. Accordingly, the authors should consider the following:

-Several studies pool early-onset PE with late-onset PE, whereas others, such as Bahado-Shing et al. (reference 24), focus exclusively on early-onset PE. This increases the heterogeneity of the review and makes it difficult to interpret the results as a whole. The authors should explicitly address this issue in both the analysis of the results and the discussion.

-The data synthesis also does not distinguish between study design, biological matrices, analytical methodology, or timing of sample collection. The review combines all these dimensions without a sufficiently differentiated structure, which weakens the conclusions. If the aim of the review is to evaluate the use of metabolomics for the prediction of preeclampsia, it is critical to consider the gestational trimester in which measurements are taken. Placental studies are conducted postpartum, and therefore contribute limited information on the predictive value of metabolomics. Early predictive biomarkers are mixed with late diagnostic biomarkers, making it difficult to draw any firm conclusions about the early predictive value of the reported metabolites.

-The manuscript concludes that there is convergence in four metabolites (alanine, lactate, glutamate, glutamine), yet these metabolites are reported as increased in some contexts and decreased in others. The authors should provide a more cautious narrative synthesis of these common metabolites and clearly indicate when each metabolite is increased, decreased, or unchanged according to sample type and analytical technique.

-Explaining the methodological heterogeneity between metabolomic technologies is essential. The review does not discuss how he different techniques (NMR, LC–MS, HR-MAS, etc) may yield different metabolite profiles. The level of confidence with which each study identifies its metabolites should be rigorously evaluated. This would help distinguish high-confidence signals from those that may correspond to isomers or analytical artefacts and would prevent treating as common metabolites those findings that might instead reflect methodological or analytical limitations.

6. PLOS authors have the option to publish the peer review history of their article (what does this mean? ). If published, this will include your full peer review and any attached files.). If published, this will include your full peer review and any attached files.

**Do you want your identity to be public for this peer review?** For information about this choice, including consent withdrawal, please see our For information about this choice, including consent withdrawal, please see our Privacy Policy ..

Reviewer #1: No

Reviewer #2: **Yes:** María T. LlinásMaría T. Llinás

---

## [Author Response · Author response to Decision Letter 1]

27 Jan 2026

Reviewer #1

1) Although the authors state that a metaanalysis is not possible due to differing methods between the studies, an extended analysis on the clinical utility of metabolomics in preeclampsia is warranted, especially limited to MS or NMR metabolomics.

RESPONSE TO THE REVIEWER: We have added a discussion on the clinical utility of metabolomics in preeclampsia, focusing on MS and NMR techniques. We explain how these methods could help with early risk assessment, disease stratification, and monitoring of PE, while noting limitations due to differences in study design, gestational age, PE subtype, and analytical methods. Examples from other diseases, such as breast cancer and multiple sclerosis, are mentioned now to show the potential of these techniques in clinical research.

2) Figure 2 attempts to summarise findings on the same metabolites between studies, but ignore differences in direction of change between studies. Would it be possible to include the direction of change, maybe as a color code? The figure also conflates the different study matrices like placenta and serum/plasma. It would be nice to see in the figure which matrix is discussed when referring to a metabolite.

RESPONSE TO THE REVIEWER: Done. Figure 2 has been ammended, and now both matrices and direction of change are included as color codes.

3) Similarly, when describing each study, a note of timing of pregnancy would benefit interpretability.

RESPONSE TO THE REVIEWER: We have now included the timing of pregnancy for all studies in which this information was not previously cited in the text.

4) Authors cite differing metabolic states of patients as a reason for the differences between studies, but do not discuss methodological and preprocessing differences between studies, for example normalization strategies and quantification methods. The review would benefit from a short discussion on these limitations as well.

RESPONSE TO THE REVIEWER: We have added a short discussion highlighting that methodological and data preprocessing differences across studies may contribute to variability in reported metabolomic findings.

5) Given the heterogeneity of findings in the studies, what is a way forward for the field. Validation of previous findings, larger exploratory studies, more specific and reproducible methodology? please discuss

RESPONSE TO THE REVIEWER: We agree with the reviewer that the heterogeneity of findings across studies represents a major challenge for the field. We have now expanded the Discussion section to explicitly address possible ways forward. In particular, we emphasize the need for: (i) stratification of study cohorts according to preeclampsia subtypes (e.g., early-onset vs. late-onset PE), (ii) validation of previously reported biomarkers in independent and adequately powered cohorts, and (iii) harmonization of pre-analytical procedures, sample collection protocols, and analytical workflows to improve reproducibility across studies. We also discuss the importance of combining exploratory metabolomic studies with targeted, hypothesis-driven approaches to enhance the translational and clinical relevance of metabolomic findings in preeclampsia.

6) Line 57, EO and LO PE are described as “usually occurring”, “normally presents” before/after 34 weeks. This is the actual definition of EO/LO PE so there is no usuall/normally, these always occur in their defined GA periods.

RESPONSE TO THE REVIEWER: Done. These expressions have been removed from the text.

7) Preeclampsia is written in full several times even though an abbreviation is provided at the start. Examples: PE defined line 48, preeclampsia spelled out line line 75, 102, 222,340 etc.

RESPONSE TO THE REVIEWER: Done. “PE” is now used consistently throughout the manuscript.

8) Weeks of gestation defined on line 340 as wkGA but then not used again, we consider appropriate to retain it in the manuscript.

RESPONSE TO THE REVIEWER: The abbreviation wkGA is indeed used later in the manuscript (lines 343–344) to refer to weeks of gestation. Therefore, we consider it appropriate to retain this abbreviation in the manuscript for consistency and clarity.

9) Gestational hypertension defined as GHT on line 258 (not actually defined in the paper) and as GH on line 267.

RESPONSE TO THE REVIEWER: Gestational hypertension has been defined now in the introduction as GH. Other abbreviations referring to this condition have been changed to GH.

10) Pregnant controls are inconsistently defined as PC (357), “Healthy controls”(394), “normotensive pregnancies”, “normal pregnancies”, “healthy women”. Although different nomenclature might reflect different populations, the review would improve readability with consistent naming of non-pregnant women, pregnant women without HDP, and PE/GH/HDP populations.

RESPONSE TO THE REVIEWER: We have reviewed all terminology to ensure it accurately reflects the study populations. Non-pregnant women are referred to as NP, and pregnant women without hypertensive disorders of pregnancy are referred to as pregnant controls (PC). In the single case where the original article defined PC differently (not as women without HDP), we have retained the terminology from the original study to accurately reflect that population. Affected groups are designated as PE, GH, or HDP, as appropriate. All abbreviations are defined at first mention and applied uniformly across the manuscript, tables, and figures.

11) PLSDA defined several times: (221,255, 293). PCA mentioned at line 377, but not at first mention (328).

RESPONSE TO THE REVIEWER: Done. Only one definition of PLS-DA has been retained, with the repeated definitions removed, and the definition of PCA has been added at its first mention.

12) Some metabolites are inconsistently named: L-glutamine, L-histidine (362, 359), L-valine (390) valine (439) glutamine, histidine (374). Similarly, please refer to metabolites by a consistent name: citrate (367) is the same metabolite as ascorbate (416). Accordingly, does this change your analysis if you did not realize they were the same?

RESPONSE TO THE REVIEWER: Done. We thank the reviewer for pointing out the inconsistent metabolite nomenclature. We have revised the manuscript to ensure consistent naming throughout (L-glutamine, L-histidine and L-valine are now used uniformly). Regarding citrate and ascorbate, we clarify that these are chemically and biologically distinct metabolites derived from citric acid and ascorbic acid, respectively. They were therefore intentionally treated as separate metabolites in our analysis. This clarification does not affect the analysis or the interpretation of the results.

13) Authors refer to results from logistic regression studies both as AUC (area under curve) (213, 243, 272) and ROC curve area (315, 330), please stay consistent.

RESPONSE TO THE REVIEWER: Done. We agree that consistent terminology is important. Accordingly, all instances previously referred to as “ROC” have now been updated to AUC throughout the manuscript.

14) FGR not defined (line 432)

RESPONSE TO THE REVIEWER: Done. Definition included.

15) Authors suggest that differences in metabolites between studies might be due to metabolic profiles of the patients (458), this is a circular argument: differences in metabolites are due to differences in metabolites. Please refine your argument

RESPONSE TO THE REVIEWER: We agree that the original statement was circular and did not sufficiently advance the interpretation of inter-study variability. Accordingly, this argument has been removed, and the Discussion has been restructured to focus instead on well-defined sources of heterogeneity, including differences in study design, patient stratification, clinical characteristics, and analytical methodologies.

16) Wording on line 70 is confusing, please rephrase “PlGF just can predict less 90% of early PE cases”. Also do you mean EO-PE as defined earlier?

RESPONSE TO THE REVIEWER: The wording on line 70 has been re`pphrased as sugested. It now correctly indicates that PlGF predicts less than 90% of early-onset PE (EO-PE) cases, as previously defined. The reference has also been corrected, as the original one was incorrect.

17) Paragraph on line 84 is not very relevant, describing cancer and Covid-19. It can be shortened to a few sentences. Perhaps a better use of this space is to provide an overview of other more or less validated biomarkers of preeclampsia, for example sFLT, PlGF and PAPP-A? This would help when these are discussed later in line 338.

RESPONSE TO THE REVIEWER: The paragraph on line 84 has been shortened as suggested, and an overview of other commonly used biomarkers for preeclampsia, including sFlt-1, PlGF, and PAPP-A, has been added. For clarity and flow, this information has been included in line 67.

18) Table 1, column “Age, average years” is unclear. What do the two numbers mean in row 1,3,6 etc. Only last row has a measure of confidence, are these standard deviance?

RESPONSE TO THE REVIEWER: All data in that column has been corrected. Now it is presented as “Maternal age, (mean ± SD or median [range])”, providing a clear measure.

19) Line 221, extra period after citation

RESPONSE TO THE REVIEWER: Done.

20) Uterine doppler PI mentioned line 246 but defined with abbreviation at line 383

RESPONSE TO THE REVIEWER: Done. For clarity and consistency, we have standardized the terminology throughout the manuscript to uterine artery pulsatility index (UtAPI), and the abbreviation has been defined at its first mention.

21) Line 273 refers to an AUC of 0, is this correct?

RESPONSE TO THE REVIEWER: No, this was not correct. The AUC value is 0.177, and this has been corrected in the text.

22) Throughout the description of the included studies, a lot of metabolites are listed. These are repeated in the table 2. Would it be possible to reduce the mentioned metabolites in text to a few most interesting ones, to improve readability? For example lines 303-310.

RESPONSE TO THE REVIEWER: We have now highlighted the most representative metabolite classes (fatty acids, acylcarnitines, and amino acid–related compounds) and all individual metabolites remain available in Table 3.

23) Table 2, abbreviations not defined in table heading/footer.

RESPONSE TO THE REVIEWER: The corresponding definitions for all abbreviations have now been added to the footnote of the tables.

24) Seems unnecessary to define abbreviations that are not used again, lines 296-299.

RESPONSE TO THE REVIEWER: The abbreviations have been removed, as the article that mentioned them has been excluded from the study.

25) Line 408, does this refer to healthy pregnant women or non pregnant women?

RESPONSE TO THE REVIEWER: It refers to healthy pregnant women, it has been changed in the text.

26) Line 430, unclear grammar. Should it be “contains almost all”?

RESPONSE TO THE REVIEWER: The sentence has been revised for clarity, and it now reads “which contained almost all LO-PE pregnancies…”.

Reviewer #2

1) The PICO question is not fully adhered to in the final study selection. The article by Liu et al. (reference 30) does not compare the group of women with preeclampsia to women with healthy pregnancies. The authors should either exclude it from the review or justify its inclusion in the text and explain the implications of this choice.

RESPONSE TO THE REVIEWER: Although the study by Liu et al. (reference 30) included women with healthy pregnancies, the metabolomic comparisons were primarily performed between women with preeclampsia and those with gestational hypertension, rather than between preeclampsia and normotensive pregnancies. As this does not strictly adhere to the predefined PICO criteria of the review, the study has now been excluded from the final analysis. It has also been removed from the corresponding tables and figures and is now listed in the appendix of excluded studies.

2) The authors do not conduct any subgroup analyses despite the evident heterogeneity among the included studies. Instead, they restrict their approach to presenting individual descriptions of each study. Accordingly, the authors should consider the following: Several studies pool early-onset PE with late-onset PE, whereas others, such as Bahado-Shing et al. (reference 24), focus exclusively on early-onset PE. This increases the heterogeneity of the review and makes it difficult to interpret the results as a whole. The authors should explicitly address this issue in both the analysis of the results and the discussion.

RESPONSE TO THE REVIEWER: The type of preeclampsia evaluated in each study (early-onset, late-onset, mixed, or not specified) is now reported in the Participants Characteristics section and has been included in Table 2. In addition, we have added a sentence in the Results section explicitly acknowledging this source of clinical heterogeneity. This issue is further discussed in the Discussion section.

3) The data synthesis also does not distinguish between study design, biological matrices, analytical methodology, or timing of sample collection. The review combines all these dimensions without a sufficiently differentiated structure, which weakens the conclusions. If the aim of the review is to evaluate the use of metabolomics for the prediction of preeclampsia, it is critical to consider the gestational trimester in which measurements are taken. Placental studies are conducted postpartum, and therefore contribute limited information on the predictive value of metabolomics. Early predictive biomarkers are mixed with late diagnostic biomarkers, making it difficult to draw any firm conclusions about the early predictive value of the reported metabolites.

RESPONSE TO THE REVIEWER: Study design, biological matrices, analytical methodology, and timing of sample collection are clearly reported in the Results tables. Additionally, the results are structured by biological matrix, and for each study, the analytical technique and sample collection timing are now explicitly indicated. Studies conducted in placental tissue have been considered in the Discussion, and we have added a clarifying sentence noting that the metabolites identified in placenta largely reflect the metabolic state at delivery and are therefore descriptive of PE at that time. Importantly, some of these metabolites were also detected in maternal serum and plasma during pregnancy, suggesting that they may have potential as early predictive biomarkers.

4) The manuscript concludes that there is convergence in four metabolites (alanine, lactate, glutamate, glutamine), yet these metabolites are reported as increased in some contexts and decreased in others. The authors should provide a more cautious narrative synthesis of these common metabolites and clearly indicate when each metabolite is increased, decreased, or unchanged according to sample type and analytical technique.

RESPONSE TO THE REVIEWER: The direction of change for each metabolite (whether increased or decreased) is now clearly indicated in all instances, with explicit mention of the sample type. In addition, we have now added the analytical technique used for each measurement to provide full context. Furthermore, Figure 2 has been updated to reflect these changes. It now not only summarizes the metabolites but also indicates whether their levels increase or decrease and specifies the sample matrix analyzed.

5) Explaining the methodological heterogeneity between metabolomic technologies is essential. The review does not discuss how he different techniques (NMR, LC–MS, HR-MAS, etc) may yield different metabolite profiles. The level of confidence with which each study identifies its metabolites should be rigorously evaluated. This would help distinguish high-confidence signals from those that may correspond to isomers or analytical artefacts and would prevent treating as common metabolites those findings that might instead reflect methodological or analytical limitations.

RESPONSE TO THE REVIEWER: We have substantially expanded the Study Characteristics section to include a dedicated analysis of the metabolomic platforms used across the selected studies, explicitly discussing their analytical principles, strengths, and limitations. This compar

---

## [Decision Letter · Decision Letter 1]

25 Feb 2026

PONE-D-25-46421R1Untargeted Metabolomics for the Early Detection of Preeclampsia: A Systematic Review of Human StudiesPLOS One

Dear Dr. Fernandez,

Thank you for submitting your manuscript to PLOS ONE. After careful consideration, we feel that it has merit but does not fully meet PLOS ONE’s publication criteria as it currently stands. Therefore, we invite you to submit a revised version of the manuscript that addresses the points raised during the review process.

The manuscript has improved substantially after revision. However, important concerns remain regarding the internal coherence of the Discussion, particularly in relation to the temporal and pathophysiological interpretation of the findings and the strength of the conclusions. We therefore ask the authors to carefully address the remaining points raised by Reviewer #2 to ensure greater interpretative clarity and alignment between the evidence presented and the conclusions drawn.

We look forward to receiving your revised manuscript.

Kind regards,

María Teresa Llinás

Academic Editor

PLOS One

Journal Requirements:

Reviewers' comments:

Reviewer's Responses to Questions

**Comments to the Author**

1. If the authors have adequately addressed your comments raised in a previous round of review and you feel that this manuscript is now acceptable for publication, you may indicate that here to bypass the “Comments to the Author” section, enter your conflict of interest statement in the “Confidential to Editor” section, and submit your "Accept" recommendation.

Reviewer #1: All comments have been addressed

Reviewer #2: (No Response)

2. Is the manuscript technically sound, and do the data support the conclusions?

Reviewer #1: Yes

Reviewer #2: No

3. Has the statistical analysis been performed appropriately and rigorously? 

Reviewer #1: N/A

Reviewer #2: N/A

4. Have the authors made all data underlying the findings in their manuscript fully available?

Reviewer #1: Yes

Reviewer #2: Yes

5. Is the manuscript presented in an intelligible fashion and written in standard English?

Reviewer #1: Yes

Reviewer #2: Yes

6. Review Comments to the Author

Reviewer #1: Dear Authors,

All comments have been satisfactorily adressed. The manuscript is now clearer and easier to follow, with more attention paid to whether metabolites were up- or downregulated and from which biological matrix and timepoint they were measured.

Reviewer #2: The manuscript has improved compared to the previous version; however, the Discussion still contains important conceptual and interpretative inconsistencies that require further revision before the work can be considered for publication. In particular, the interpretative structure of the Discussion still lacks full temporal and pathophysiological coherence, especially regarding the directionality of metabolite changes and their biological context. While heterogeneity is now acknowledged and more explicitly described in tables and text, it is not yet fully integrated into a structured analytical interpretation. As a result, some mechanistic explanations remain overly generalized or partially speculative, and the strength of the conclusions may still exceed what is supported by the available evidence.

First, the interpretation of alanine discrepancies remains overly generic. Attributing opposite directions of change to “analytical techniques” and “clinical heterogeneity” is insufficient without a more explicit mechanistic rationale. The discussion should distinguish between systemic versus placental metabolism and between early adaptive metabolic shifts and later-stage dysfunction. Without this clarification, the explanatory model remains incomplete. Moreover, this limitation also weakens the strength of the overall conclusions of the review, as the proposed convergence of metabolites cannot be adequately supported if the direction and underlying biological context of these changes are not coherently interpreted.

Second, the explanation of lactate findings remains physiologically inconsistent. The reported decrease in lactate was observed in first trimester samples from women who subsequently developed late-onset PE. Nevertheless, the discussion relies on placental hypoxia as the main mechanistic explanation. Placental hypoxia is typically associated with later stage or early onset PE rather than first trimester preclinical LO-PE, and hypoxia-driven anaerobic glycolysis would be expected to increase lactate production, not decrease it. As currently written, the proposed mechanism does not adequately account for the observed direction of change. The authors should resolve this discrepancy and provide a pathophysiological explanation that is consistent with the specific gestational window studied rather than extrapolating mechanisms from later stages of disease.

With respect to the section of the Discussion addressing glutamate and glutamine, clearer compartmental and temporal differentiation is needed. Serum and placental findings are presented within a unified framework without sufficiently distinguishing systemic maternal metabolism from local placental processes. I would recommend restructuring this section to clearly distinguish systemic maternal findings from placental tissue results and to integrate a temporal perspective (early predictive vs late descriptive changes).

Finally, the concluding statement suggesting that the current evidence provides a “solid basis” for future validation appears overly optimistic, given the limited number of studies, the inconsistency in the direction of metabolite changes, and the absence of quantitative synthesis. A more cautious and proportionate tone would be advisable to better reflect the strength of the available evidence.

7. PLOS authors have the option to publish the peer review history of their article (what does this mean? ). If published, this will include your full peer review and any attached files.). If published, this will include your full peer review and any attached files.

**Do you want your identity to be public for this peer review?** For information about this choice, including consent withdrawal, please see our For information about this choice, including consent withdrawal, please see our Privacy Policy ..

Reviewer #1: No

Reviewer #2: No

---

## [Author Response · Author response to Decision Letter 2]

1 Mar 2026

Prof. María Teresa Llinás

Academic Editor PLOS One

Dear Editor,

We have revised the manuscript as requested, addressing all the concerns detailed below. Additionally, we are uploading a file with all the changes highlighted in red.

Reviewer #2

We sincerely thank the reviewer for the careful and constructive evaluation of our manuscript.

1) First, the interpretation of alanine discrepancies remains overly generic. Attributing opposite directions of change to “analytical techniques” and “clinical heterogeneity” is insufficient without a more explicit mechanistic rationale. The discussion should distinguish between systemic versus placental metabolism and between early adaptive metabolic shifts and later-stage dysfunction. Without this clarification, the explanatory model remains incomplete. Moreover, this limitation also weakens the strength of the overall conclusions of the review, as the proposed convergence of metabolites cannot be adequately supported if the direction and underlying biological context of these changes are not coherently interpreted.

RESPONSE TO THE REVIEWER: We agree with the reviewer that temporal stage and biological compartment are essential for interpreting metabolomic findings in preeclampsia. To address this, we have added a new opening paragraph to the Discussion explicitly introducing a framework that distinguishes: (i) early predictive maternal signals vs. late descriptive placental changes, and (ii) systemic maternal metabolism vs. local placental metabolism. This paragraph clarifies that apparent inconsistencies in metabolite directionality are expected when observations derive from different physiological phases or compartments rather than representing purely analytical discrepancies. This conceptual framing now guides the interpretation of subsequent metabolite-specific sections.

Regarding the interpretation of alanine discrepancies, we have substantially rewritten the alanine section. The revised text now: i) distinguishes maternal systemic metabolism from placental metabolism; ii) separates early adaptive metabolic responses from late-stage dysfunction; and iii) proposes a mechanistic interpretation linking alanine dynamics to altered hepatic gluconeogenesis, nitrogen transport, and systemic metabolic adaptation in early pregnancy. This restructuring provides a physiologically grounded explanation for opposite directions of change across studies and matrices.

2) Second, the explanation of lactate findings remains physiologically inconsistent. The reported decrease in lactate was observed in first trimester samples from women who subsequently developed late-onset PE. Nevertheless, the discussion relies on placental hypoxia as the main mechanistic explanation. Placental hypoxia is typically associated with later stage or early onset PE rather than first trimester preclinical LO-PE, and hypoxia-driven anaerobic glycolysis would be expected to increase lactate production, not decrease it. As currently written, the proposed mechanism does not adequately account for the observed direction of change. The authors should resolve this discrepancy and provide a pathophysiological explanation that is consistent with the specific gestational window studied rather than extrapolating mechanisms from later stages of disease.

RESPONSE TO THE REVIEWER: We agree that attributing first-trimester lactate reductions to placental hypoxia was not appropriate. The lactate section has therefore been revised to distinguish early systemic metabolic adaptation from later hypoxia-driven placental dysfunction. The revised discussion now proposes that reduced early lactate may reflect altered maternal metabolic flexibility or gluconeogenic regulation rather than hypoxia, which is more relevant to established disease stages. This modification aligns the mechanistic interpretation with the gestational window studied.

3) With respect to the section of the Discussion addressing glutamate and glutamine, clearer compartmental and temporal differentiation is needed. Serum and placental findings are presented within a unified framework without sufficiently distinguishing systemic maternal metabolism from local placental processes. I would recommend restructuring this section to clearly distinguish systemic maternal findings from placental tissue results and to integrate a temporal perspective (early predictive vs late descriptive changes).

RESPONSE TO THE REVIEWER: We have restructured this section to explicitly separate maternal biofluid findings from placental tissue observations and to distinguish predictive signals from descriptive late-stage metabolic alterations. The revised text now integrates these metabolites within systemic redox balance, immune regulation, and nitrogen metabolism, while clearly acknowledging the temporal context of each observation.

4) Finally, the concluding statement suggesting that the current evidence provides a “solid basis” for future validation appears overly optimistic, given the limited number of studies, the inconsistency in the direction of metabolite changes, and the absence of quantitative synthesis. A more cautious and proportionate tone would be advisable to better reflect the strength of the available evidence.

RESPONSE TO THE REVIEWER: We agree and have moderated the tone of the concluding paragraph. The revised conclusion now explicitly acknowledges the limited number of studies, variability in gestational timing, PE subtype definitions, and analytical platforms, as well as the inconsistent directionality observed for some metabolites. Rather than presenting metabolomics as providing definitive predictive biomarkers, the revised text emphasizes that the current literature primarily identifies plausible metabolic pathways and physiological axes that warrant further investigation. This change ensures that the conclusions remain proportionate to the strength of the available evidence.

We hope that all these changes will be satisfactory, and we are uploading the revised version of the manuscript We are confident that the changes done will meet with your approval.

Many thanks in advance.

Sincerely,

Ignacio Fernández

---

## [Decision Letter · Decision Letter 2]

8 Mar 2026

PONE-D-25-46421R2Untargeted Metabolomics for the Early Detection of Preeclampsia: A Systematic Review of Human StudiesPLOS One

Dear Dr. Fernandez,

Thank you for submitting your manuscript to PLOS ONE. After careful consideration, we feel that it has merit but does not fully meet PLOS ONE’s publication criteria as it currently stands. Therefore, we invite you to submit a revised version of the manuscript that addresses the points raised during the review process.

The manuscript has been positively evaluated by the reviewers and has improved substantially following revision. However, several minor revisions are still required in the Discussion to improve clarity and internal coherence. The authors should carefully address these minor recommendations in the revised version.

We look forward to receiving your revised manuscript.

Kind regards,

María Teresa Llinás

Academic Editor

PLOS One

Journal Requirements:

Reviewers' comments:

Reviewer's Responses to Questions

**Comments to the Author**

1. If the authors have adequately addressed your comments raised in a previous round of review and you feel that this manuscript is now acceptable for publication, you may indicate that here to bypass the “Comments to the Author” section, enter your conflict of interest statement in the “Confidential to Editor” section, and submit your "Accept" recommendation.

Reviewer #2: All comments have been addressed

2. Is the manuscript technically sound, and do the data support the conclusions?

Reviewer #2: Yes

3. Has the statistical analysis been performed appropriately and rigorously? 

Reviewer #2: Yes

4. Have the authors made all data underlying the findings in their manuscript fully available?

Reviewer #2: Yes

5. Is the manuscript presented in an intelligible fashion and written in standard English?

Reviewer #2: Yes

6. Review Comments to the Author

Reviewer #2: Although the Discussion has clearly improved compared with the previous version, a few minor adjustments would further strengthen its internal coherence. In particular, the interpretation of alanine and lactate would benefit from a more integrated explanation within the pyruvate metabolic framework. The paragraph describing pyruvate metabolism (lines 535-542) under hypoxic conditions introduces mechanisms that would favor increased lactate production and potentially increased alanine formation, whereas the study discussed reports decreased levels of both metabolites in first-trimester maternal samples. Clarifying that the hypoxia-related metabolic routing is more characteristic of later stages of established disease and considering alternative explanations for the simultaneous decrease (e.g., increased hepatic utilization of lactate and alanine as gluconeogenic substrates, enhanced oxidative utilization of pyruvate, or subtype-specific metabolic patterns such as in late-onset PE), would improve internal consistency.

In addition, the discussion of glutamate and glutamine could be more explicitly integrated with the previously described pyruvate–alanine–lactate metabolic axis. Presenting these metabolites within a unified metabolic network, rather than as separate pathways, would strengthen the physiological coherence of the discussion and help reconcile the differing directions reported across studies.

More generally, the Discussion could further emphasize the interpretative contribution of the review. Beyond compiling the available studies, a review should ideally provide a coherent synthesis that helps explain how the reported metabolic alterations may be biologically connected. Strengthening this integrative perspective would enhance the added value of the review and place the individual findings within a clearer physiological context.

Finally, two paragraphs convey very similar ideas regarding compartment- and stage-dependent metabolic differences (lines 498–502 and 547–556). Streamlining or merging these passages would help reduce redundancy and improve clarity

7. PLOS authors have the option to publish the peer review history of their article (what does this mean? ). If published, this will include your full peer review and any attached files.). If published, this will include your full peer review and any attached files.

**Do you want your identity to be public for this peer review?** For information about this choice, including consent withdrawal, please see our For information about this choice, including consent withdrawal, please see our Privacy Policy ..

Reviewer #2: **Yes:** María T. LlinásMaría T. Llinás

---

## [Author Response · Author response to Decision Letter 3]

9 Mar 2026

Dear Editor,

We have revised the manuscript as requested, addressing all the concerns detailed below. Additionally, we are uploading a file with all the changes highlighted in red.

Reviewer #2

We sincerely thank the reviewer for the careful evaluation of our manuscript and for the constructive suggestions that have helped us improve the internal coherence and interpretative strength of the Discussion section. Below we address each point in detail.

1) The interpretation of alanine and lactate would benefit from a more integrated explanation within the pyruvate metabolic framework… The study discussed reports decreased levels of both metabolites in first-trimester maternal samples… consider alternative explanations…

RESPONSE TO THE REVIEWER: We have clarified that the hypoxia-driven metabolic routing described is more characteristic of later stages of established disease rather than the early gestational window captured in first-trimester maternal metabolomic studies. In addition, we have incorporated alternative physiological explanations for the simultaneous decrease in lactate and alanine, including increased hepatic utilization of both metabolites as gluconeogenic substrates, enhanced mitochondrial oxidative use of pyruvate, and metabolic heterogeneity across PE subtypes. These additions reconciles apparently divergent findings and improve the internal consistency of the Discussion.

2) The discussion of glutamate and glutamine could be more explicitly integrated with the previously described pyruvate–alanine–lactate metabolic axis.

RESPONSE TO THE REVIEWER: We agree and have revised the text to present glutamate and glutamine metabolism within an integrated metabolic network. As suggested, we now emphasize how transamination reactions functionally connect amino acid metabolism with central carbon metabolism, linking the glutamate–glutamine axis to the pyruvate–alanine–lactate pathway.

3) The Discussion could further emphasize the interpretative contribution of the review… provide a coherent synthesis…

RESPONSE TO THE REVIEWER: We appreciate this important suggestion. To strengthen the integrative perspective, we have revised the concluding section to emphasize the interpretative value of the review. The revised text highlights how apparently distinct metabolite alterations converge into interconnected metabolic networks and coherent physiological axes, reinforcing the added value of the review beyond summarizing individual studies.

4) Two paragraphs convey very similar ideas regarding compartment- and stage-dependent metabolic differences… Streamlining or merging would help reduce redundancy…

RESPONSE TO THE REVIEWER: We agree and have streamlined the Discussion accordingly. The earlier general paragraph describing compartment- and stage-dependent differences has been retained, while the later paragraph has been condensed to avoid conceptual redundancy and improve clarity and flow.

We thank the reviewer again for these valuable suggestions, and we hope that all these changes will be satisfactory. We are uploading the revised version of the manuscript and are confident that the changes done will meet with your approval.

Many thanks in advance.

Sincerely,

Ignacio Fernández

---

## [Editor Report · Decision Letter 3]

15 Mar 2026

Untargeted Metabolomics for the Early Detection of Preeclampsia: A Systematic Review of Human Studies

PONE-D-25-46421R3

Dear Dr. Fernandez,

We’re pleased to inform you that your manuscript has been judged scientifically suitable for publication and will be formally accepted for publication once it meets all outstanding technical requirements.

Kind regards,

María Teresa Llinás

Academic Editor

PLOS One
---

## [Editor Report · Acceptance letter]

PONE-D-25-46421R3

PLOS One

Dear Dr. Fernandez,

I'm pleased to inform you that your manuscript has been deemed suitable for publication in PLOS One. Congratulations! Your manuscript is now being handed over to our production team.

Kind regards,

on behalf of

Dr. María Teresa Llinás

Academic Editor

PLOS One